# Host Response of Human Epidermis to Methicillin-Resistant *Staphylococcus aureus* Biofilm Infection and Synthetic Antibiofilm Peptide Treatment

**DOI:** 10.3390/cells11213459

**Published:** 2022-11-01

**Authors:** Bing (Catherine) Wu, Travis M. Blimkie, Evan F. Haney, Reza Falsafi, Noushin Akhoundsadegh, Robert E. W. Hancock

**Affiliations:** Centre for Microbial Diseases and Immunity Research, Department of Microbiology and Immunology, University of British Columbia, Vancouver, BC V6T1Z4, Canada

**Keywords:** biofilm immunopathogenesis, innate immune response, inflammation, skin barrier function, extracellular matrix remodeling, DNA repair, host defense peptide, human skin, RNA-Seq transcriptomic analysis

## Abstract

Bacterial biofilm infections associated with wounded skin are prevalent, recalcitrant, and in urgent need of treatments. Additionally, host responses in the skin to biofilm infections are not well understood. Here we employed a human organoid skin model to explore the transcriptomic changes of thermally-injured epidermis to methicillin-resistant *Staphylococcus aureus* (MRSA) biofilm colonization. MRSA biofilm impaired skin barrier function, enhanced extracellular matrix remodelling, elicited inflammatory responses including IL-17, IL-12 family and IL-6 family interleukin signalling, and modulated skin metabolism. Synthetic antibiofilm peptide DJK-5 effectively diminished MRSA biofilm and associated skin inflammation in wounded human ex vivo skin. In the epidermis, DJK-5 shifted the overall skin transcriptome towards homeostasis including modulating the biofilm induced inflammatory response, promoting the skin DNA repair function, and downregulating MRSA invasion of thermally damaged skin. These data clarified the underlying immunopathogenesis of biofilm infections and revealed the intrinsic promise of synthetic peptides in reducing inflammation and biofilm infections.

## 1. Introduction

The emergence of antibiotic resistance poses a global public health concern leading to prolonged illness, increased treatment failure, and elevated rates of disability and mortality [1]. *Staphylococcus aureus,* ranked highly important in the World Health Organization’s global priority list of antibiotic-resistant bacterial pathogens [2], can form multicellular biofilms that are 10- to 1000-times more resistant to conventional antibiotics when compared to planktonic bacteria [3]. Critically, *S. aureus* is a common cause of skin and soft tissue infections [4] and the most frequently isolated bacterium from chronic wound infections (e.g., found in 88–93.5% of chronic venous leg ulcers), causing excessive inflammation and delaying wound healing and the re-epithelialization process [5]. In one study, all 160 *S. aureus* isolates from patients with skin infections were capable of biofilm formation, suggesting that biofilms are a universal behavior of *S. aureus* skin infections [6].

Human skin, the largest organ of the human body, functions as a physical barrier between the external environment and internal organs, provides immune protection, regulates body temperature and water balance, and supplies hormones and neurotransmitters [7]. The host response to biofilm infections is complex and dynamic, since bacterial signature molecules that are readily recognized by the host immune system can be hindered by extracellular polymeric substances and other components of the biofilm extracellular matrix as it develops [8]. This is particularly complicated for *S. aureus* infections since this organism is capable of both intracellular and extracellular survival [9]. The inflammatory response also depends on the site of infection and types of immune cells present. Immune responses against *S. aureus* infections involve coordinated activities of immune cells (e.g., neutrophils, macrophages, B cells, and T cells) and host defense peptides [10]. Stratified keratinocytes in the epidermis are essential for skin barrier function, structural integrity, and initiation of skin inflammation. However, their role in biofilm infection is not well understood.

Effective treatments for skin biofilm infections are urgently required, since the clinical pipeline of new antimicrobials is currently dry and there is currently no treatment specifically targeting biofilm infections. Host defense peptides (HDPs), also known as antimicrobial peptides, are short, positively charged polypeptides that are rich in hydrophobic residues [11]. Synthetic analogs of naturally occurring HDPs have improved beneficial activities such as antibiofilm as well as immunomodulatory and anti-inflammatory properties while maintaining low cytotoxicity toward mammalian cells [12]. For example, DJK- 5, a D-enantiomeric peptide, has broad-spectrum activity in both biofilm inhibition and eradication [13,14]. As a potentially universal mechanism of action, DJK-5 targets the bacterial stringent response, required for biofilm formation, by binding and triggering the degradation of the stringent response alarmones guanosine tetraphosphate and pentaphosphate [15]. As a promising anti-biofilm therapeutic candidate, it is of interest to explore the influence of DJK-5 on host tissue responses during biofilm infections.

In this study we found that methicillin-resistant *Staphylococcus aureus* (MRSA) biofilm infection suppressed genes involved in cornified envelope formation such as loricrin and keratins. Additionally, biofilm infection increased both skin extracellular matrix (ECM) formation and degradation and enhanced the expression of matrix metalloproteinases (MMPs) that mediate tissue remodeling during physiological or pathological processes. DJK-5 treatment modulated MRSA biofilm-induced skin inflammation, especially the interleukin response, by downregulating IL-12 and IL-6 family signalling while promoting IL-10, IL-4, and IL-13 cascades. DJK-5 also uniquely promoted pathways regulating cell cycle progression and the repair of DNA damage promoting the DNA damage sensor, ATR, and the BRCA1-associated genome surveillance complex components expression, which play roles in DNA mismatch repair, excisional repair, and the homologous recombination pathway. DJK-5 reduced MRSA intracellular invasion, in part by downregulating MRSA interaction with host cell adhesion molecules, intracellular kinase signalling cascades and MRSA mobilization of actin cytoskeleton.

## 2. Material and Methods 

### 2.1. Peptide and Reagents

Peptide DJK-5 (*VQWRAIRVRVIR-*NH_2_; all D amino acids as indicated in italics) was synthesized by CPC Scientific (Sunnyvale, CA, USA) using solid-phase 9-fluorenylmethoxy carbonyl chemistry and purified to ~95% using reverse-phase high-performance liquid chromatography. Peptide identity was confirmed by mass spectrometry. N/TERT cell and skin culture medium including Keratinocyte-SFM medium, Dulbecco’s Modified Eagle Medium (DMEM, high glucose, GlutaMAX™ Supplement, pyruvate), and Ham’s F-12 Nutrient Mix were purchased from ThermoFisher Scientific (Waltham, MA, USA). CnT-Prime 3D Barrier Medium was obtained from CELLnTEC Advanced Cell Systems AG (Zurich, Switzerland). DermaLife K Keratinocyte Medium Complete Kit was purchased from Lifeline Cell Technology (Oceanside, CA, USA). Supplements for skin culture medium including isoproterenol, hydrocortisone, bovine insulin, selenious acid, l-serine, l-carnitine, bovine serum albumin, palmitic acid, linoleic acid, and arachidonic acid were obtained from Sigma-Aldrich (St. Louis, MI, USA). Antibiotic controls (gentamicin and fusidic acid) and 10% neutral-buffered formalin solution were also purchased from Sigma-Aldrich. Reagents for bacterial culture and resuspension including tryptic soy broth (TSB), Luria broth, d-glucose, and phosphate-buffered saline (PBS) were purchased from ThermoFisher Scientific.

### 2.2. Bacteria Culture 

MRSA USA300-LAC [16] was grown overnight in TSB (supplemented with 1% d-glucose) at 37 °C with shaking at 180 rpm. The next day, bacteria were sub-cultured and grown to mid-exponential growth phase in TSB (1% d-glucose), harvested by centrifugation at 6200× *g* for 5 min and resuspended in PBS or TSB containing 1% d-glucose before seeding 2 × 10^6^ CFU onto each skin sample.

### 2.3. MRSA USA300-LAC Thermal Wounding ex Vivo Skin Model

The surplus human skin experimental protocol (H18-000657) was approved by the UBC Clinical Research Ethics Board and the College of Physicians and Surgeons of British Columbia. Healthy breast surplus skin samples (3–5 mm in thickness) were collected from consenting healthy donors (age 19–45) post-breast reduction surgery. The procedure for ex vivo skin culture and infection was adapted from de Breij et al. [17]. Skin was rinsed 3 times with PBS and cut into 8 mm biopsies. The apical side of the skin was thermally injured with a digital soldering iron (FX888D, American Hakko Products Inc., Valencia, CA, USA) at 150 °C for 10 s. A reservoir for infection and treatment was created by building a barrier surrounding the skin containing the burn wound with a light-curing dental liquid dam (Ultradent Products Inc., South Jordan, UT, USA), followed by 4 s of ultraviolet light fixation. Skin with treatment reservoir was cultured in a 12-well plate at the air–liquid interface on a metal wire rack with 3.2 mL of the culture medium (DMEM/Ham’s F-12/CnT-Prime 3D Barrier Medium in a 3:1:4 ratio supplemented with 0.1 μg/mL hydrocortisone, 0.125 μg/mL isoproterenol, 0.25 μg/mL bovine insulin, 26.5 pM selenious acid, 5 mM L-serine, 5 μM L-carnitine, 1.6 mg/mL BSA, 25 μM palmitic acid, 30 μM linoleic acid, and 7 μM arachidonic acid) underneath. MRSA biofilm was established by seeding 3 μL of 6.7 × 10^8^ CFU/mL (2 × 10^6^ CFU) USA300-LAC in TSB (1% d-glucose) on top of the thermally damaged skin and cultured at 37 °C and 5% CO_2_ for 24 h. Skin biofilm was treated with 5 μL of 0.1% (5 μg) or 0.4% (20 μg) DJK-5 or fusidic acid dissolved in water or 5 μL of water as a control for 4 h. To quantify bacterial counts, each skin sample was homogenized using a mini-beadbeater-96 (BioSpec Products Inc., Bartlesville, OK, USA) for 3 cycles of 30 s in 1 mL PBS with a 3 mm diameter Tungsten carbide bead (Qiagen, Hilden, Germany), vortexed, serially diluted, and plated on Luria broth agar plates. The colony count detection limit was 10 CFU/skin. To determine the concentrations of cytokines and chemokine released by the ex vivo skin samples, culture medium underneath the skin was used to measure IL-1β, IL-6, and IL-8 production using ELISA kits from eBioscience.

### 2.4. H&E Staining

Ex vivo skin samples were sandwiched between two foam biopsy pads (ThermoFisher Scientific) in a tissue embedding cassette (Sigma-Aldrich) and fixed in 10% neutral-buffered formalin for 48 h. H&E staining of skin and biofilm cross sections was performed by Wax-it Histology Services Inc. (Vancouver, BC, Canada) and images were analyzed using the Aperio ImageScope software v12.4.0.5043 (Leica Biosystems, Wetzlar, Germany).

### 2.5. MRSA USA300-LAC Thermal Wounding N/TERT Skin Model

N/TERT keratinocytes were provided by Dr. Peter Nibbering (Leiden University Medical Center) with permission from Dr. James Rheinwald (Harvard Medical School). N/TERT cell culture and the N/TERT skin biofilm model were carried out as published previously [18]. In brief, skin models were created from 3 × 10^5^ N/TERT cells on filter inserts (ThinCert™ Cell culture insert, Greiner bio-one) in a 12-well ThinCert™ Plate (Greiner bio-one) as described previously. When cells reached confluency (after 3–4 days) the culture medium was switched to a differentiation medium (DMEM/Ham’s F-12/CnT-Prime 3D Barrier Media in a 3:1:4 ratio supplemented with 0.1 μg/mL hydrocortisone, 0.125 μg/mL isoproterenol, 0.25 μg/mL bovine insulin, 26.5 pM selenious acid, 5 mM L-serine, 5 μM L-carnitine, 1.6 mg/mL BSA, 25 μM palmitic acid, 15 μM linoleic acid, and 7 μM arachidonic acid). The apical side of the skin was air-exposed the next day to induce stratification. After 2–3 days, the concentration of linoleic acid in the differentiation medium was increased to 30 μM. N/TERT skin was ready for experiments after culturing at the air–liquid interface at 37 °C and 7.3% CO_2_ for 10 days. Thermal damage was created by applying a digital soldering iron to N/TERT skin at 100 °C for 4 s. MRSA biofilm was established by seeding 5 μL of 4 × 10^8^ CFU/mL (2 × 10^6^ CFU) on top of the thermally damaged skin and cultured at 37 °C and 5% CO_2_ for 24 h. DJK-5 peptide (30 μL of 0.4%) was administered on top of the pre-formed biofilm. Skin samples were collected 24 h post-peptide treatment for RNA-isolation. 

### 2.6. RNA Isolation 

N/TERT epidermal skin was collected from 4 treatment groups, namely: (1) untreated skin control; (2) burned skin control (skin thermally challenged at 100 °C for 4 s); (3) burned skin with MRSA biofilm (burned skin spotted with 2 × 10^6^ CFU MRSA USA300-LAC for 24 h then treated with 30 µL water for 24 h); and (4) burned skin with MRSA biofilm and DJK-5 treatment (burned skin with 24 h MRSA USA300-LAC biofilm then treated with 30 μL of 0.4% DJK-5 for 24 h). Skin models were excised from the cell culture insert using a disposable scalpel (VWR, Radnor, PA, USA) and immediately submerged in 800 µL RNA*later* RNA stabilization solution (ThermoFisher Scientific), stored at 4 °C overnight and then transferred to a −80 °C freezer until the RNA isolation could be performed. To harvest enough RNA for RNA-Seq analysis, three skin models with the same treatment were pooled into each sample before RNA extraction. Total RNA was extracted from four independent pooled samples per treatment group using the RNeasy Micro Kit (Qiagen) following the manufacturer’s protocol. For quality control, 1 µL of each sample was run on the Agilent 2100 Bioanalyzer using the Eukaryotic Total RNA Nano Chip (Agilent Technologies, Santa Clara, CA, USA). 

### 2.7. RNA-Seq Library Preparation 

To ensure data quality, the changes in expression of representative inflammatory genes were confirmed using RT-qPCR (Appendix A). The primer sequences used in the RT-qPCR experiments are listed in Appendix A. The production of inflammatory cytokine IL-1β and chemokine IL-8 from the skin culture medium were measured using ELISA (Appendix A). 

RNA-Seq library construction was performed as described previously [19] using 1–2 µg of each RNA sample. Poly-A tailed RNA enrichment was done using the Magnetic mRNA Isolation Kit (New England Biolabs, Ipswich, MA, USA). To prepare complementary DNA libraries, mRNAs were enzymatically fragmented followed by first and second strand complementary DNA synthesis and unique indices were ligated using the Kapa Stranded RNA-Seq Kit (Kapa Biosystems, Wilmington, MA, USA). DNA libraries were amplified by polymerase chain reaction followed by cleaning and size selection using the AMPure XP kit (Agencourt Bioscience, Beverly, MA, USA). DNA samples were quantified using the Quant-iT™ dsDNA Assay Kit (ThermoFisher Scientific) and normalized to 4 nM. RNA-Seq libraries were sequenced on a HiSeqX sequencer (Illumina) at Canada’s Michael Smith Genome Sciences Centre at the British Columbia Cancer Agency.

### 2.8. RNA-Seq Analysis

Quality control of the N/TERT skin sequenced data was performed using FastQC [20] v0.11.8 and MultiQC v1.8 [21]. Sample libraries were then aligned to the human reference genome, Ensembl GRCh38 v98 [22] using STAR v2.7.3a [23]. Uniquely mapped reads included a minimum of 12.3 million, median of 29.5 million, and a maximum of 47.6 million reads. A read count table was generated using HTSeq-count v0.11.2 [24], and genes that had fewer than 50 counts across four biological replicates were removed. Differentially expressed (DE) gene analysis was performed using DESeq2 v1.28.1 with the Wald statistical test, and DE genes were considered significant if they had an absolute fold change value of ≥1.5 and adjusted *p*-value ≤ 0.05 [25]. Reactome [26] pathway enrichment of DE genes was performed using ReactomePA [27], with significance defined as a Bonferroni-adjusted *p*-value ≤ 0.05 [28]. Network analysis was done by uploading genes and their respective fold change values to NetworkAnalyst for construction of protein–protein interaction networks [29]. 

### 2.9. Statistical Analysis 

Statistical significance of bacterial colony counts recovered from the ex vivo skin model was determined by the Kruskal–Wallis test (a non-parametric test) with Dunn’s multiple comparisons. Statistical significance of cytokine and chemokine concentrations from the ex vivo skin comparing peptide treated skins to MRSA control was performed using one-way ANOVA, Dunnett’s multiple comparisons test. Statistical analysis was performed using GraphPad Prism Version 8.0.2. Statistical significance was reported using the following cut-offs: * *p*-value ≤ 0.05; ** *p*-value ≤ 0.01; *** *p*-value ≤ 0.001; **** *p*-value ≤ 0.0001.

## 3. Results 

### 3.1. DJK-5 Reduced MRSA Biofilm and Inflammation in Thermally Wounded Human Skin 

We previously showed that DJK-5 effectively killed MRSA in biofilms grown on thermally-injured epidermal skin organoids and dampened biofilm-induced skin inflammation [18]. To study if the antibiofilm and anti-inflammatory activities of DJK-5 were conserved under conditions more representative of human burn wounds, we employed an air–liquid interface model using human breast surplus skin (Figure 1). Unlike the N/TERT skin, this model contained, in addition to epidermis, both dermis and immune cells. H&E staining of a human skin cross section showed this structural complexity with the epidermal rete ridges extending into the dermis (Figure 1a). To facilitate biofilm colonization, the surface of this ex vivo skin was thermally injured at 150 °C for 10 s before MRSA infection (Figure 1b). This thermal injury caused a large number of epidermal cells to lose their circular appearance and severely damaged the integration of the epidermis, causing it to separate from the dermis, while no major alterations in the dermis were observed. Consistent with previous observations with N/TERT epidermal skin [18], one-day MRSA biofilm appeared as purple clusters associated with the epidermis (Figure 1c), and the remaining biofilms or debris were stained pink after peptide treatments (Figure 1d). Colony counts recovered after 4 h peptide treatment of MRSA biofilm showed that DJK-5 reduced the CFU by about 2 and 5 log orders of magnitude at 0.1% (5 μg) and 0.4% (20 μg) respectively (Figure 1e). Notably, 0.4% DJK-5 performed considerably better than the antibiotic control, 0.4% fusidic acid, by around 3 log orders of magnitude. Additionally, DJK-5 at both doses completely inhibited IL-1β production (Figure 1f), and 0.4% DJK-5 significantly suppressed IL-6 (Figure 1g) and IL-8 (Figure 1h) levels by about 3.4-fold and 4.1-fold respectively. 

### 3.2. Effect of DJK-5 Treatment on MRSA Biofilm Infected Thermally-Injured Skin Transcriptome

Keratinocytes are the predominant cells in the epidermis. They recognize microbial pathogens in part through Toll-like receptors (TLRs), and act as the first line of innate immune defense against infection [30]. Since histological changes were prominently observed in the epidermis (Figure 1a–d), we employed the epidermal skin organoid model to further study the impact of MRSA infection and 0.4% (120 ng) DJK-5 treatment on the host epidermal immune response. 

RNA-Seq analysis was performed on total mRNA samples extracted from burned skin controls and thermally-challenged N/TERT epidermal skin infected with 24 h MRSA biofilm, followed by 24 h DJK-5 treatment. When comparing other treatment groups to burned skin, genes were considered differentially expressed (DE), if they had ≥1.5 absolute fold changes, with an adjusted *p*-value < 0.05. MRSA biofilm infection of burned skin led to very large transcriptomic changes with 3674 upregulated genes and 3700 downregulated genes when compared to the burned skin control (Appendix A). DJK-5-treated, MRSA-infected burned skin also had a large number of DE genes when compared to MRSA biofilm infected skin (3411 upregulated and 3396 downregulated). This appeared to be largely due to the reversal of the inflammation and skin damage caused by MRSA biofilm infection, since comparing DJK-5-treated, MRSA-infected burned skin to the burned control revealed very few DE genes (152 upregulated and 7 downregulated, Appendix A). Thus DJK-5 treatment shifted the overall skin transcriptome closer to homeostasis without infection. 

### 3.3. MRSA Biofilm Infection Impaired Skin Barrier Function and Enhanced Extracellular Matrix Turnover in Thermally-Injured Skin 

The cornified envelope, layers of terminally differentiated keratinocytes atop the skin, functions as a permeability and mechanical defense barrier preventing pathogen colonization through its low water content, acidic pH, associated normal microflora, and surface-deposited antimicrobial lipids [31]. Pathway enrichment analysis using ReactomePA [27] revealed the differential influence of MRSA infection and DJK-5 treatment on the burned skin (Table 1). MRSA biofilm infection impaired the formation of cornified envelope (adjusted *p* < 1.1 × 10^−2^). NetworkAnalyst provides a method to display functional interactions (direct, biochemical, and regulatory) of dysregulated genes as a protein–protein interaction network. While using this method, it was discovered that MRSA biofilm infection downregulated the expression of loricrin and multiple components of the keratin intermediate filaments (e.g., KRT1, 2, 9 and 10), which are responsible for forming strong tissue networks and providing strength and resiliency to the skin [32] (Figure 2a). DJK-5 treatment upregulated this pathway (1.3 × 10^−2^), indicating a protective role in restoring barrier function (Table 1). 

The host ECM is a highly dynamic yet strictly regulated tissue component that plays essential roles during immune response to infections (e.g., microbial recognition, and immune cell recruitment and activation) and regulation of inflammatory networks [33]. MRSA biofilm upregulated pathways in cell-extracellular matrix interactions (2.2 × 10^−4^) and both the formation and degradation of ECM, which were correspondingly downregulated by DJK-5 treatment (Table 1). These genes include type IV, type VII, and type XVI collagens, laminins (e.g., LAMA1, 3, 5, LAMB1, 3 and LAMC1, 2), integrins (e.g., ITGA1, 2, 3, 5, 6 and ITGB1, 4, 6, 7) and matrix metalloproteinases (e.g., MMP1, 3 and 9) (Figure 2b), reflecting increased ECM remodeling processes in response to MRSA colonization and skin inflammation [34].

### 3.4. DJK-5 Dampened MRSA Biofilm-Induced Skin Inflammation 

In the burned skin, MRSA biofilm infection provoked innate immune responses, including pathways directing proinflammatory cell recruitment (*p* = 3.5 × 10^−2^), TLR cascades (1.9 × 10^−3^), MAP kinase activation (1.9 × 10^−3^), and several signal transduction pathways (Table 1). In addition, multiple cytokine pathways such as IL-17 (3.2 × 10^−3^), IL-12 family (3.5 × 10^−2^), and IL-6 family (7.7 × 10^−3^) were upregulated by MRSA biofilm infection. These included skin inflammation related cytokines such as IL17C (fold-change: 803.8), IL23A (68.3), LIF (10.6), IL11 (5.2), MIF (2.2), and essential transcription factors such as AP-1 subunits FOS (31.9) and JUN (6.9), NFKB1 (3.0), ATF1 (1.6), ATF2 (3.0), and CREB1 (2.4) (Figure 3). Skin treated with DJK-5 led to an overall dampened inflammatory state by downregulating the above pathways. In particular, DJK-5 modulated the interleukin response by downregulating IL-12 (*p* = 8.6 × 10^−3^) and IL-6 (8.6 × 10^−3^) family signalling (compared to MRSA biofilm infected skin) while promoting IL-10 (5.2 × 10^−6^), IL-4 and IL-13 (2.2 × 10^−3^) signalling (compared to the burned skin control) (Table 1). 

### 3.5. DJK-5 Reduced the Cellular Stress Response to Amino Acid Starvation

Host integrated stress response can shape the innate immune response during infections through detection of cellular stresses and damages caused by pathogenic bacteria regardless of their specific virulence factors [35]. MRSA biofilm infection increased skin metabolism including rRNA processing (*p* = 4.3 × 10^−9^), translation (2.1 × 10^−4^), and metabolism of amino acids (1.1 × 10^−2^), reflecting increased protein synthesis demands for host defenses (e.g., synthesis of proinflammatory cytokines) (Table 1). MRSA skin biofilm treated with DJK-5 resulted in reduced RNA, protein and amino acid metabolism. The decreased demand for amino acids also contributed to the reduction in cellular stress responses including the cellular response to starvation (*p* = 5.4 × 10^−21^) and the response of protein kinase EIF2AK4 sensing amino acid deficiency (1.2 × 10^−36^) (Table 1), which is mediated by binding to uncharged tRNAs near the ribosome and phosphorylating the translation initiation factor EIF2, resulting in decreased translation [36].

### 3.6. DJK-5 Promoted DNA Repair Function in MRSA Biofilm Infected Skin

DNA repair mechanisms are essential host responses in correcting physico-chemical aberrations in the genome and can arouse the immune system through, for example, activation of immune signalling pathways and induction of antimicrobial peptide expression [37]. Pathway enrichment analysis (Table 1) revealed that DJK-5 treatment uniquely promoted pathways regulating cell cycle progression and the repair of DNA damage, therefore we further investigated DE genes within these pathways using known protein–protein interactions as visualized by NetworkAnalyst (Figure 4). Compared to MRSA biofilm infected skin without peptide treatment, skin treated with DJK-5 had elevated expression of DNA damage sensor ATR serine/threonine kinase and multiple components of the BRCA1-associated genome surveillance complex including BRCA1, BLM, NBN, and PCNA and subunits of the replicative factor C (e.g., RFC2, RFC3, and RFC5) that are involved in DNA mismatch repair and excisional repair [38]. In addition, peptide treatment also upregulated the expression of key factors in the homologous recombination pathway for double-strand DNA repair (e.g., PALB2, BRCA2, RPA1, RPA2, RAD51D, RAD51B, RAD51AP1, and XRCC2) [39]. Together these results suggested that DJK-5 was able to reverse effects on the recognition and repair of aberrant DNA structures during MRSA biofilm skin infection.

### 3.7. DJK-5 Reduced Pathways Mediating MRSA Invasion of Thermally-Injured Skin 

Studies have demonstrated that MRSA can invade and persist within host cells including keratinocytes [40,41]. Therefore, we further studied the effect of DJK-5 on bacterial internalization upon colonization. MRSA infection of thermally wounded skin upregulated multiple skin extracellular matrix genes such as collagen, fibronectin, and laminin as well as their integrin receptors (Figure 5). Following fibronectin and integrin-mediated adhesion, MRSA infection upregulated multiple kinases such as the integrin-linked kinase (ILK), focal adhesion kinase (FAK), and Src kinase. Additionally, activation of the hepatocyte growth factor receptor MET led to upregulation of downstream WAVE family proteins, which associated with an actin related protein 2/3 complex and function to enhance actin polymerization and internalization of MRSA. Conversely, DJK-5 treatment of biofilm infected burned skin led to downregulation of both the fibronectin–integrin pathway and the MET pathway including the essential kinases (e.g., ILK, Src and PI3K), and resulted in reduced septin assembly. These results indicated that DJK-5 reduced the mechanisms driving MRSA intracellular invasion.

## 4. Discussion

Skin is an essential protective barrier of the human body that is constantly challenged by environmental insults and microbial pathogens and is considered to have an important role in early immune responses [42]. Biofilms colonizing wounds can impede wound healing [43] and cause hyper-inflammatory [44] responses that are detrimental to the host. In particular, *S. aureus* biofilms are among the most common in burn and chronic wounds [45]. Despite the high prevalence and severe consequences, there is currently limited knowledge about the impact of *S. aureus* biofilm on essential biological functions of human skin during infection. 

The results of the ex vivo human skin model confirmed the antibiofilm and anti-inflammatory effects of DJK-5 against MRSA biofilm (Figure 1). These results were consistent with our previous study using the epidermal N/TERT skin model, where DJK-5 was able to reduce the numbers of live biofilm bacteria by >2000-fold, MRSA-induced IL-1β secretion by 10-fold, and IL-8 secretion by 2.4-fold [18]. This wounded surplus skin model, as compared to the N/TERT epidermal skin model [18], added structural complexity (e.g., dermis) and cellular diversity (e.g., CD14^+^ or CD1c^+^ monocytes/macrophages, CD11c^+^ DCs, CD56^+^CD3^-^ natural killer cells, CD4^+^ T cells, CD8^+^ T cells, and CD19^+^ B cells [46,47]), and better reflected donor variability. However, the complexity of the ex vivo skin model due to genetic differences in individual donors and its compositional complexity, makes molecular studies difficult since any changes observed are hard to ascribe to any given cell type. The use here of an organoid model based on differentiated keratinocytes overcame these issues and was justified by the similar morphology (of the epidermis) and response to MRSA and DJK-5 peptide.

Using transcriptomics coupled with bioinformatic analysis, we explored skin organoid epithelial responses to MRSA biofilm infection and DJK-5 treatment. Although H&E staining showed substantial skin damage in the epidermis due to thermal injury [18], surprisingly no DE genes were found when comparing burned skin to untreated skin (Appendix A). This is likely because the burned skin controls (as for other experimental conditions) were collected 48 h post-thermal challenge for RNA-Seq analysis, and this was likely too late to capture transcriptomic changes due to moderate thermal injury. By comparison, MRSA biofilm infection triggered tremendous transcriptional changes in burned skin. For example, MRSA biofilm suppressed skin structural genes such as loricrin and keratins that are responsible for maintaining the skin barrier function and the mechanical stability of individual cells in the epidermal tissue (Figure 2a), while breaching such defenses, renders skin tissue more vulnerable to functionally diverse virulence factors (e.g., toxins and immune modulators) produced by MRSA [48]. The disruption of skin integrity, observed histologically, also helps to explain the refractory nature of *S. aureus* biofilm associated with chronic inflammatory conditions such as in the atopic dermatitis skin lesions [49,50]. Skin extracellular matrix (ECM) is a three-dimensional scaffold interwoven with multiple components that provides physical strength and elasticity, facilitates immune cell recruitment and regulates inflammation in response to pathologic stimuli [34]. Upon infection, pathogens can alter the synthesis and turnover of ECM and impact on its composition and spatial distribution [33]. For example, Rhinovirus infections induce airway remodeling through increased deposition of ECM components such as fibronectin, perlecan, and collagen IV [51]. 

Here we observed that MRSA biofilm infection increased the expression of skin basement membrane-related ECM components including type IV collagen, type VII collagen, laminins and their integrin receptors (Figure 2b). MRSA biofilm also increased the expression of MMPs (e.g., MMP1, 3 and 9), a class of proteinases involved in extracellular matrix degradation, potentially affecting the ability of MRSA to colonize and invade skin tissue. Importantly, MMPs are associated with pathological processes of chronic skin inflammation. MMP3 and MMP9 have been found to accumulate in psoriatic plaques, cause cleavage of basement membrane, and facilitate pathogenic T cell infiltration [52]. MMP9 induction also contributes to the activation of IL-1β and plasminogen, mediates skin damage, and impairs wound healing [52]. Similarly in biofilm wound infection, the consequential excessive release of harmful MMPs prolongs the inflammatory response and fuels biofilm formation [53]. Indeed, MRSA biofilm elicited multiple interleukin cascades, including the strong induction of IL-17C and IL23A expression (Figure 3). Previous studies have shown that induction of IL-17A and IL-17F from γδ T cells is essential for neutrophil recruitment and host defenses against cutaneous *S. aureus* infection in mice [54,55] while IL-17C has been shown to promote wound closure during *S. aureus* wound infection in mice [56]. Critically, the IL-17 and IL-23 axis have well-recognized roles in chronic inflammation [57]; IL-23 stimulates IL-17 production by activating Th17 cells and elevated expression of IL-23 has been observed in psoriatic lesional skin [57,58]. IL-17C amplifies epithelial inflammation in human psoriasis and atopic eczema by potentiating the expression of other cytokines, chemokines, and HDPs as well as the autocrine induction of IL-17C in keratinocytes [59]. Interestingly, MRSA biofilm also induced the expression of the IL-6 family cytokines LIF and IL-11, both of which exert anti-inflammatory effects such as favoring regulatory T cell development, promoting alternative macrophage differentiation, and reducing systemic TNF production in mouse models of endotoxin-induced septic shock [60,61]. Together these data highlight the complexity of host immune responses in stratified keratinocytes to MRSA biofilm infections, whereby injured skin recognizes MRSA surface antigens (e.g., lipoteichoic acid), biofilm extracellular matrix components (e.g., exopolysaccharides and extracellular DNA), and debrided dead host cells [62], both on the skin surface and after invasion into host cells. 

In addition to the direct antibiofilm activity shown in Figure 1, DJK-5 treatment also modulated host skin transcriptional responses so that they became similar to the burned skin control without MRSA infection, including promoting cornified envelope formation and reducing ECM turnover (Table 1). Interestingly DJK-5 modulated the inflammatory response by promoting anti-inflammatory IL-10 and Th-2 cytokine IL-4 and IL-13 signalling without significantly affecting IL-17 cascades. Similarly, other synthetic HDPs, such as IDR-1018, have been shown to drive macrophage differentiation towards an intermediate M1-M2 state, enhancing anti-inflammatory functions while maintaining certain pro-inflammatory activities important to the resolution of infections [63]. This shifting in immune response is beneficial as prolonged inflammation and reactive oxygen species generated by bacterial infection often result in genomic instability. For example, *S. aureus* infections can trigger DNA damage and delay the cell cycle transition through induction of oxidative stress [64,65]. Skin cancer-associated *S. aureus* secretomes have also been shown to suppress DNA repair mechanisms including homologous recombination, mismatch, base excision, nucleotide excision, and double-strand break repair in keratinocytes [65]. Here we reported a novel protective function of synthetic host defense peptide DJK-5 through upregulation of DNA repair mechanisms (Figure 4). DJK-5 promoted the expression of ATR, which responds to a broad spectrum of DNA damage (e.g., double-stranded DNA breaks and stalled replication forks caused by DNA lesions) and regulates cell cycle progression and the repair of DNA damage [66]. In addition, DJK-5 enhanced the expression of DNA repair machineries, such as components of the BRCA1-associated genome surveillance complex, which can alter composition depending on the type of DNA damage and the cell cycle status [38]. For example, DJK-5 enhanced BRCA1 together with BRCA2 and RAD51, which direct DNA double-strand break repair through homologous recombination. PALB2 and RAD51AP1 have been shown to stimulate RAD51 recombinase activity and D-loop formation [67]. The interaction of PALB2 and BRCA1 might fine-tune the location of BRCA2-RAD51 repair machinery at DNA breaks [68]. DJK-5 also upregulated genes that participated in the base excision repair pathway, a major DNA repair pathway that protects mammalian cells against single-base DNA damage by genotoxicants such as methylating and oxidizing agents. This pathway included POLD, POLE which catalyze DNA strand displacement synthesis (resolution of abasic sites) during the S-phase of the cell cycle in the presence of PCNA, RPA, RFC, FEN1, and LIG1 [69]. In addition, DJK-5 exhibited a protective role that would prevent intracellular invasion by MRSA. DJK-5 treatment suppressed MRSA biofilm-induced fibronectin and integrin expression in N/TERT epidermal skin (Figure 5). Interactions between *S. aureus* fibronectin-binding proteins (e.g., FnBPA and FnBPB) and host fibronectin and α5β1 integrins mediate bacterial adhesion and invasion of non-professional phagocytic cells such as epithelial cells, endothelial cells, and fibroblasts [70]. Keratinocytes have been shown to increase the expression of fibronectin and α5β1 integrins upon wounding [71], providing more potential sites for MRSA attachment. DJK-5 treatment also downregulated several downstream kinases such as Src, ILK, and PI3K that are involved in *S. aureus* invasion of host cells [72,73,74]. Therefore, DJK-5 treatment would be able to reduce the susceptibility of thermally damaged skin to MRSA infection, in part by reducing bacterial adhesion and invasion. Interestingly, DJK-5 also suppressed the expression of the receptor tyrosine kinase MET, which navigates internalization of *Listeria monocytogenes* upon binding of the bacterial surface protein InlB [75], suggesting a potential protective role against invasion of other bacterial species. 

## 5. Conclusions

In conclusion, synthetic host defense peptide DJK-5 had promising antibiofilm and anti-inflammatory effects in MRSA biofilm infected burned human skin. Using system biology approaches, we studied the influence of MRSA biofilm and the treatment with DJK-5 on the skin transcriptome including effects on skin barrier function, ECM remodeling, and immune responses including inflammatory mechanisms, therefore providing crucial data on the dynamic host responses of epidermal skin to biofilm infections (Fold-change of representative genes in each biological function was summarized in Appendix A). Thus, skin organoid infection serves as an excellent model for teasing apart mechanisms of *S. aureus* pathogenesis. These results also revealed the protective roles of DJK-5 treatment, beyond directly targeting MRSA biofilm, such as restoring skin barrier function, enhancing beneficial immune reactions, promoting the sensing and repair of aberrant DNA structures, and reducing MRSA intracellular invasion.

## Figures and Tables

**Figure 1 cells-11-03459-f001:**
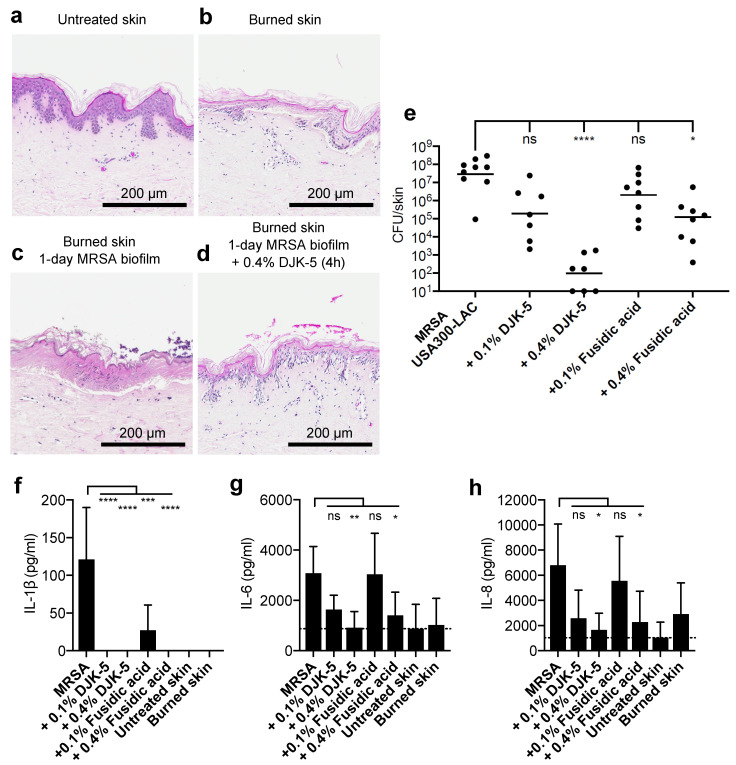
Antibiofilm activity of DJK-5 in the human burned skin MRSA biofilm model. Human surplus skin obtained post-breast reduction surgery was thermally challenged at 150 °C for 10 s and cultured at the air–liquid interface. Two million MRSA (USA300-LAC) were spotted on top of the burned skin and cultured for 24 h, followed by 4 h topical peptide treatment [0.1% (5 µg) or 0.4% (20 µg) DJK-5 or fusidic acid]. Skin cross sections were visualized by H&E staining (**a**–**d**). The colony count recovered from each skin sample was determined (**e**). Statistical significance comparing peptide treated skins to MRSA control was performed using the Kruskal–Wallis test, Dunn’s multiple comparisons test. Geometric mean of colony-count from 3 donors (7–9 replicates per conditions) was indicated. The concentrations of IL-1β (**f**), IL-6 (**g**), and IL-8 (**h**) in the skin culture medium was determined by ELISA. Error bars indicate mean with standard error in (**f**–**h**). Statistical significance of 5 replicates per condition from 3 donors was performed using one-way ANOVA, Dunnett’s multiple comparisons test (* *p* ≤ 0.05; ** *p* ≤ 0.01; *** *p* ≤ 0.001; **** *p* ≤ 0.0001).

**Figure 2 cells-11-03459-f002:**
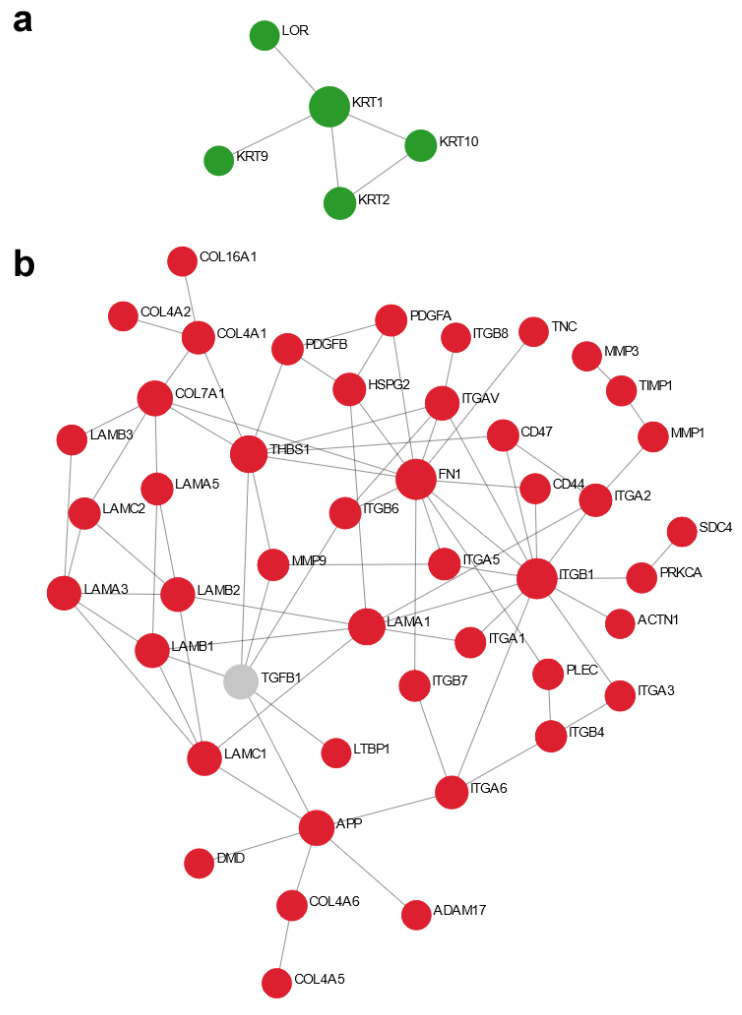
NetworkAnalyst depiction of modulation of skin barrier function and extracellular matrix turnover by MRSA biofilm infection in thermally-injured skin. Minimum protein–protein interaction networks of cornified envelope formation (**a**) and extracellular matrix organization (**b**) comparing MRSA biofilm infected skin with burned skin control using NetworkAnalyst. Red nodes denote upregulation. Green nodes denote downregulation.

**Figure 3 cells-11-03459-f003:**
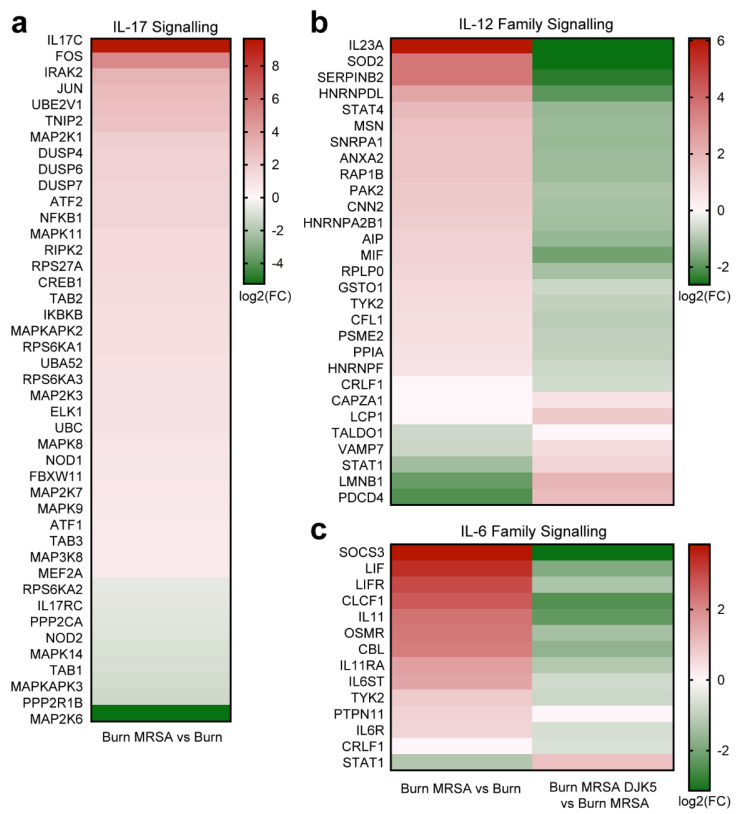
DJK-5 treatment downregulated IL-12 and IL-6 family interleukin responses in MRSA biofilm skin infection. Heatmaps of DE genes involved in the (**a**) IL-17, (**b**) IL-12 family (**c**), and IL-6 family signalling cascades in response to MRSA biofilm-induced skin inflammation with or without DJK-5 treatment. Color scale based on log_2_fold-change of DE genes with red indicating upregulation and green indicating downregulation.

**Figure 4 cells-11-03459-f004:**
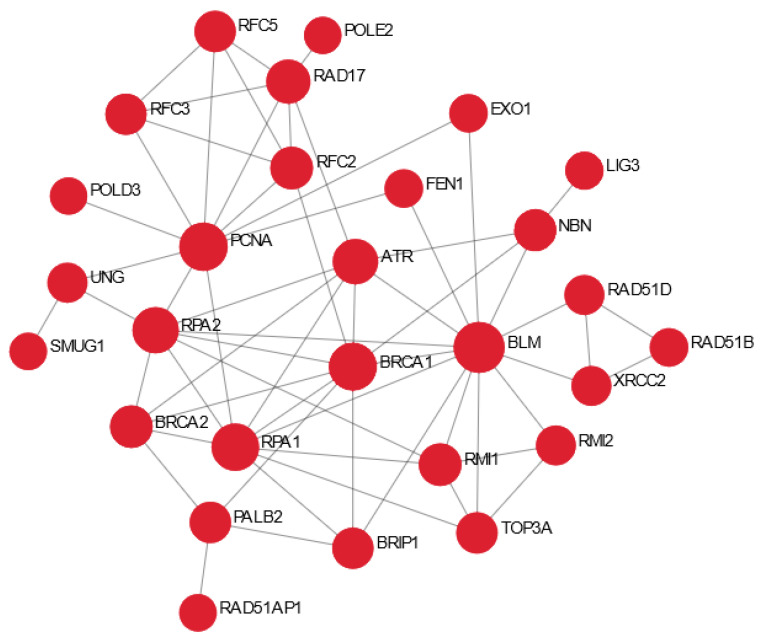
Network indicating DJK-5 enhancement of DNA repair functions in biofilm infected skin. A zero-order protein–protein interaction network of DE genes comparing DJK-5 treated vs. untreated MRSA biofilm skin infection using NetworkAnalyst. Red nodes denote upregulation.

**Figure 5 cells-11-03459-f005:**
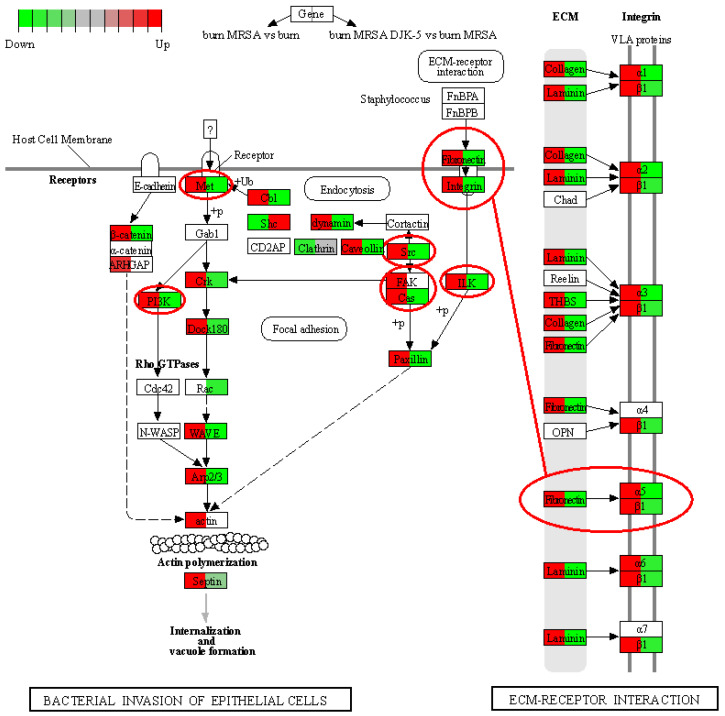
DJK-5 downregulated MRSA invasion of thermally damaged N/TERT skin. Modified KEGG Pathview graph of extracellular matrix–receptor interactions and bacterial invasion of epithelial cells. Left half of each gene compares MRSA infected burned skin to burned skin control, and the right half of each gene compares DJK-5 treated to untreated MRSA infection of burned skin. Red indicates upregulation and green indicates downregulation.

**Table 1 cells-11-03459-t001:** Selected pathways dysregulated by MRSA biofilm infection with or without DJK-5 treatment in thermally damaged N/TERT skin. Pathway enrichment analysis was performed on DE genes using ReactomePA [27] with an adjusted *p*-value < 0.05 used as the cut-off for overrepresented differentially expressed pathways.

Function	Pathway	Direction	Adjusted *p*-Value
**MRSA biofilm infected burned skin vs. burned skin control**			
Keratinization	Formation of the cornified envelope	down	1.1 × 10^−2^
Cell junction organization	Cell-extracellular matrix interactions	up	2.2 × 10^−4^
Extracellular matrix organization	Anchoring fibril formation	up	7.2 × 10^−3^
	Crosslinking of collagen fibrils	up	3.5 × 10^−2^
	Degradation of the extracellular matrix	up	8.6 ×10^−4^
	Collagen degradation	up	1.0 × 10^−2^
Infectious disease	Cell recruitment (pro-inflammatory response)	up	3.5 × 10^−2^
Innate immune system	Toll-like receptor cascades	up	1.9 × 10^−3^
	MAP kinase activation	up	1.9 × 10^−3^
Cytokine signalling in immune system	Interleukin-17 signalling	up	3.2 × 10^−3^
	Interleukin-12 family signalling	up	3.5 × 10^−2^
	Interleukin-6 family signalling	up	7.7 × 10^−3^
Signal transduction	GPCR downstream signalling	up	6.0 × 10^−8^
	Death receptor signalling	up	3.3 × 10^−3^
	RHO GTPase cycle	up	7.7 × 10^−3^
Metabolism of RNA	rRNA processing	up	4.3 × 10^−9^
Metabolism of proteins	Translation	up	2.1 × 10^−4^
	Post-translational protein phosphorylation	up	5.1 × 10^−3^
Metabolism of amino acids	Metabolism of amino acids and derivatives	up	1.1 × 10^−2^
Metabolism of lipids	Peroxisomal lipid metabolism	down	1.1 × 10^−2^
**DJK-5 treated MRSA biofilm infected burned skin vs. biofilm infected burned skin**			
Keratinization	Formation of the cornified envelope	up	1.3 × 10^−2^
Cell junction organization	Cell-extracellular matrix interactions	down	2.6 × 10^-2^
Extracellular matrix organization	Anchoring fibril formation	down	5.3 × 10^−3^
	Crosslinking of collagen fibrils	down	3.0 × 10^−2^
	Collagen degradation	down	3.7 × 10^−2^
Toll-like receptor cascades	TLR3 cascade	down	4.6 × 10^−2^
	TLR9 cascade	down	3.8 × 10^−2^
Cytokine signalling in immune system	IL-12 family signalling	down	8.6 × 10^−3^
	IL-6 family signalling	down	5.3 × 10^−3^
Signal transduction	Signalling by MET	down	8.4 × 10^−4^
	Signalling by ERBB2	down	5.1 × 10^−3^
	Death receptor signalling	down	1.2 × 10^−2^
	GPCR downstream signalling	down	1.3 × 10^−5^
Cell cycle	Activation of ATR in response to replication stress	up	4.6 × 10^−2^
DNA repair	Homologous DNA pairing and strand exchange	up	2.9 × 10^−2^
	Resolution of abasic sites	up	4.6 × 10^−2^
Metabolism	rRNA processing	down	9.5 × 10^−13^
	Translation	down	2.8 × 10^−11^
	Post-translational protein phosphorylation	down	4.7 × 10^−3^
	Metabolism of amino acids and derivatives	down	2.0 × 10^−4^
	Peroxisomal lipid metabolism	up	1.7 × 10^−2^
Cellular responses to stress	Cellular response to starvation	down	5.4 × 10^v21^
	Response of EIF2AK4 to amino acid deficiency	down	1.2 × 10^−36^
**DJK-5 treated MRSA biofilm infected burned skin vs. burned skin control**			
Cytokine signalling in immune system	IL-10 signalling	up	5.2 × 10^−6^
	IL-4 and IL-13 signalling	up	2.2 × 10^−3^
Signalling by GPCR	Chemokine receptors bind chemokines	up	4.2 × 10^−2^

## Data Availability

The RNA-Seq data presented in this study was deposited to NCBI’s Gene Expression Omnibus and is available under the accession number GSE210640 (reviewer token: kngzkeiebxcvryf).

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
