# Peer review of "Host Response of Human Epidermis to Methicillin-Resistant Staphylococcus aureus Biofilm Infection and Synthetic Antibiofilm Peptide Treatment"

_cells, 2022, doi:10.3390/cells11213459_

Round 1

Reviewer 1 Report

The manuscript examined the effect of MRSA biofilm on thermally-injured epidermis. The antibiofilm activity of DJK-5 was also evaluated. Cytokine levels were assessed and the transcriptome change was analyzed. Examination in the transcriptomic alteration is meaningful but needs further experimental validation. Currently, there is no data to prove that the suggested bioinformatic results are happening. Manual experiments to confirm some of the biomarkers are needed.

The study shows various analysis based on the RNA seq data. However, validation on these data should be present to suggest a scientifically reliable finding.

The authors suggest that MRSA and DKJ-5 affected the expression of skin barrier proteins, ECM proteins, and ECM regulating proteins. While RNA seq can be a useful tool to observe the overall change in the physiology and mechanism, key biomarkers should be confirmed manually to ensure the findings are true. Some of the representative markers for each part should be confirmed at the mRNA or/and protein level.

Similarly, some of the major cytokine (e.g. IL17C, IL23A) levels and DNA repair proteins can be measured to confirm if the data from the bulk RNA seq is actually true. 

Reviewer 2 Report

Dear authors of the paper titled "Host response to methicillin-resistance S. aureus biofilm infection and synthetic anti-biofilm peptide treatment" has several valuable scientific contents that present scientific evidence on a  peptide formulation that has the potential to attenuate the S. aureus biofilm formation in burn wounds. The introduction section is relevant, the materials and methods section provides detailed information on the methods used, the results are concisely presented, and only the results obtained in this study are appropriately discussed.

Overall the findings are exemplary. Like the authors, this reviewer is also surprised by the finding that there were no differentially expressed genes between burned skin and untreated skin. The author's explanation for this is not very convincing. This also raises questions about the analysis parameters used to compare different experimental conditions. The other primary concern this reviewer has is the lack of quantitative determination of the findings from the bioinformatic analyses. To validate the findings from the bioinformatic analyses, a handful of up-and down-regulated genes should be confirmed at the protein levels either by western blots or ELISA. In my opinion, this will add significant value to the manuscript. 

Round 2

Reviewer 1 Report

The revised manuscript did not adequately resolve the issue raised.

While RNA-seq can be a valuable tool to observe the overall change in the physiology and mechanism, some of the key genes found through analyzing the RNA-seq result should be confirmed manually to ensure the findings are true. Some of the representative genes for each part should be confirmed at the mRNA or/and protein level. The majority of the manuscript is made with data from RNA-seq. If there were other ample experimental proof to confirm the effect on skin barrier function, extracellular matrix remodelling,  inflammatory response, skin DNA repair function, etc., then the RNA-seq could be reliable by itself. However, in the current manuscript, the authors are making all the conclusions merely from the transcriptome (i.e. RNA-seq) analysis without any further confirmation or other experiments. Hence, at least some manual confirmation on the mRNA level is necessary.

Reviewer 2 Report

Dear Authors, 

Thanks for responding to the comments and for providing a detailed description of the queries raised by this reviewer. The manuscript is easy to read and understand and flows well. 

Thanks,

Author Response

Thanks for reviewing and helping to improve our manuscript!